# Robust Exploration via Clustering-based Online Density Estimation

## Abstract

Intrinsic motivation is a critical ingredient in reinforcement learning to enable progress when rewards are sparse. However, many existing approaches that measure the novelty of observations are brittle, or rely on restrictive assumptions about the environment which limit generality. We introduce *Robust Exploration via Clustering-based Online Density Estimation* (RECODE), a non-parametric method that estimates visitation counts for clusters of states that are similar according to the metric induced by a specified representation learning technique. We adapt classical clustering algorithms to the online setting to design a new type of memory that allows RECODE to efficiently track global visitation counts over thousands of episodes. RECODE can easily leverage both off-the-shelf and novel representation learning techniques. We introduce a novel generalization of the action-prediction representation that leverages transformers for multi-step predictions, which we demonstrate to be more performant on a suite of challenging 3D-exploration tasks in DM-HARD-8. We show experimentally that our approach can work with a variety of RL agents, obtaining state-of-the-art performance on Atari and DM-HARD-8, and being the first agent to reach the end-screeen in *Pitfall!*

## 1 Introduction

Exploration mechanisms are a key component of reinforcement learning (RL, Sutton & Barto, 2018) agents, especially in sparse-reward tasks where long sequences of actions need to be executed before collecting a reward. The exploration problem has been studied theoretically (Kearns & Singh, 2002; Azar et al., 2017; Brafman & Tennenholtz, 2003; Auer et al., 2002; Agrawal & Goyal, 2012; Audibert et al., 2010; Jin et al., 2020) in the context of bandits (Lattimore & Szepesvári, 2020) and Markov Decision Processes (MDP, Puterman, 1990; Jaksch et al., 2010). Among those theoretical works, one simple and theoretically-sound approach to perform exploration efficiently in MDPs is to use a decreasing function of the visitation counts as an exploration bonus (Strehl & Littman, 2008; Azar et al., 2017). However, this approach is intractable with large or continuous state spaces, as generalization between states becomes essential. Several experimental works have tried to come up with ways to estimate visitation counts/densities in complex environments where counting is not trivial. Two partially successful approaches have emerged to empirically estimate visitation counts/densities in deep RL: (i) the parametric approach that uses neural networks and (ii) the non-parametric approach that uses a slot-based memory to store representations of visited states, where the representation learning method serves to induce a more meaningful metric[1] between states.

Parametric methods either explicitly estimate the visitation counts using density models (Bellemare et al., 2016; Ostrovski et al., 2017) or implicitly estimate the counts using e.g., Random Network Distillation (RND, Burda et al., 2019; Badia et al., 2020b). Non-parametric methods rely on a memory to store encountered state representations (Badia et al., 2020b) and representation learning to construct a metric that differentiates states meaningfully (Pathak et al., 2017). Parametric methods do not store individual states explicitly and as such their capacity is not directly bound by memory constraints; but they are less well suited to rapid adaptation on short timescales (e.g., within a single episode). To obtain the best of both worlds, Never Give Up (NGU, Badia et al., 2020b) combines

---

[1] Usually this is a pseudometric on the space of observations, since $d(x, y) = 0$ for $x \neq y$ is permitted by typical network architectures, and may be desirable as a means to discard noisy or uncontrollable features

a short-term novelty signal based on an episodic memory and a long-term novelty via RND, into a single intrinsic reward. However, this approach also naturally inherits the disadvantages of RND; in particular, susceptibility to uncontrollable or noisy features (see Section 5), and being difficult to tune. More details on related works are provided in App. C.

In this paper, we propose to decompose the exploration problem into two orthogonal sub-problems. First, (i) **Representation Learning** which is the task of learning an embedding function on observations or trajectories that encodes a meaningful notion of similarity. Second, (ii) **Density Estimation** which is the task of estimating smoothed visitation counts to derive a novelty-based exploration bonus. We first present a general solution to (ii) which is computationally efficient and scalable to complex environments. We introduce Robust Exploration via Clustering-based Online Density Estimation (RECODE), a non-parametric method that estimates visitation counts for clusters of states that are similar according to a metric induced by any arbitrary representation. We adapt classical clustering algorithms to an online setting, resulting in a new type of memory that allows RECODE to keep track of histories of interactions spanning thousands of episodes. This is in contrast to existing non-parametric exploration methods, which store only the recent history and in practice usually only account for the current episode. The resulting exploration bonus is principled, simple, and matches or exceeds state-of-the-art exploration results on Atari; being the first agent to reach the end-screen in *Pitfall!*. In the presence of noise, we show that it strictly improves over state-of-the-art exploration bonuses such as NGU or RND. The generality of RECODE also allows us to easily leverage both off-the-shelf and novel representation learning techniques, which leads in to our second contribution. Specifically, we generalize the action-prediction representations (Pathak et al., 2017), used in several state-of-the-art exploration agents, by applying transformers to masked trajectories of state and action embeddings for multi-step action prediction. We refer to this method as CASM for *Coupled Action-State Masking*. In conjunction with RECODE, CASM can yield significant performance gains in hard 3D-exploration tasks included in the DM-HARD-8 suite; achieving a new state of the art in the single-task setting.

## 2 BACKGROUND AND NOTATION

In this section, we provide the necessary background and notation to understand our method (see Sec. 3). First, we present a general setting of interaction between an agent and its environment. Second, we define the terms embeddings, atoms and memory. Third, we present our notation for visitation counts. Finally, we show how we derive intrinsic rewards from visitations counts.

**Interaction Process between an Agent and its Environment.** We consider a discrete-time interaction process (McCallum, 1995; Hutter, 2004; Hutter et al., 2009; Daswani et al., 2013) between an agent and its environment where, at each time step $t \in \mathbb{N}$, the agent receives an observation $o_t \in \mathcal{O}$ and generates an action $a_t \in \mathcal{A}$. We consider an environment with stochastic dynamics $p : \mathcal{H} \times \mathcal{A} \to \Delta_{\mathcal{O}}{}^2$ that maps a history of past observations-actions and a current action to a probability distribution over future observations. More precisely, the space of past observations-actions is $\mathcal{H} = \bigcup_{t \in \mathbb{N}} \mathcal{H}_t$ where $\mathcal{H}_0 = \mathcal{O}$ and $\forall t \in \mathbb{N}^*, \mathcal{H}_{t+1} = \mathcal{H}_t \times \mathcal{A} \times \mathcal{O}$. We consider policies $\pi : \mathcal{H} \to \Delta_{\mathcal{A}}$ that maps a history of past observations-actions to a probability distribution over actions. Finally, an extrinsic reward function $r_e : \mathcal{H} \times \mathcal{A} \to \mathbb{R}$ maps a history to a scalar feedback.

**Embeddings, Atoms and Memory.** An embedder is a parameterized function $f_\theta : \mathcal{H} \to \mathcal{E}$ where $\mathcal{E}$ is an embedding space. Typically, the embedding space is the vector space $\mathbb{R}^N$ where $N \in \mathbb{N}^*$ is the embedding size. Therefore, for a given time step $t \in \mathbb{N}$, an embedder is a function $f_\theta$ that associates to any history $h_t \in \mathcal{H}_t$ a vector $e_t = f_\theta(h_t)$ called an embedding. There are several ways to train an embedder $f_\theta$ such as using an auto-encoding loss of the observation $o_t$ (Burda et al., 2018a), using an inverse dynamics loss (Pathak et al., 2017) or using a multi-step prediction-error loss at the latent level (Guo et al., 2020; 2022). Those techniques are referred as representation learning methods. An atom $f \in \mathcal{E}$ is a vector in the embedding space that is contained in a memory $M = \{f_i \in \mathcal{E}\}_{i=1}^{|M|}$ which is a finite slot-based container, where $|M| \in \mathbb{N}^*$ is the memory size. The memory $M$ is updated at each time step $t$ by a non-parametric function of the memory $M$ and the embedding $e_t$. In the simplest case, the memory is filled in a first-in first-out (FIFO) manner along

---

[2]We write $\Delta_{\mathcal{Y}}$ the set of probability distributions over a set $\mathcal{Y}$.

the interactions (Badia et al., 2020b;a) and atoms are simply the embeddings themselves. However, more complex mechanisms than FIFO can be considered to fill/update a memory. For instance given a memory $M$ and an embedding $e_t$, the embedding $e_t$ can be inserted in the memory if and only if it is sufficiently different from the other atoms in the memory and the memory is not at capacity. The update rule that defines the atoms in the memory is a key component of our method.

**Visitation Counts.** The exact visitation count, $N_\delta(M, e)$, for a given embedding $e \in \mathcal{E}$ with respect to the memory $M$ can be written as:

$$N_\delta(M, e) = \sum_{l=1}^{|M|} \delta(f_l, e), \text{ where } \delta(f, e) : (e, f) \in \mathcal{E}^2 = \begin{cases} 1, & \text{if } e = f \\ 0, & \text{otherwise,} \end{cases} \tag{1}$$

However, when the state-space is very large or continuous, the exact visitation count is often un-informative since the same embedding may rarely be encountered twice. To overcome this problem, we can instead compute *soft-visitation counts* $N_\mathcal{K}(M, e) := \sum_{l=1}^{|M|} \mathcal{K}(f_l, e)$, where $\mathcal{K} \in \mathbb{R}_+^{\mathcal{E}^2}$ is a definite positive kernel. Different choices of kernel can be made such as a Gaussian kernel $\mathcal{K}(f, e) = \exp(-\|e - f\|_2^2)$ or an inverse kernel $\mathcal{K}(f, e) = \frac{1}{1+\|e-f\|_2^2}$ where $\|.\|_2$ is the Eu-clidean distance in the embedding space $\mathcal{E}$. Finally, we can compute a weighted soft visitation count $N_\mathcal{K}(M, e, \{w_l\}_{l=1}^{|M|}) = \sum_{l=1}^{|M|} w_l \mathcal{K}(f_l, e)$, where the weights $w_l \in \mathbb{R}_+$ are positive real num-bers denoting the count at each atom. Note that the exact visitation count can be recovered from the more general weighted soft visitation count as a special case by setting the weights $w_l = 1$ and $\mathcal{K} = \delta$.

**Intrinsic Rewards from Visitation Counts.** It is straightforward to define an intrinsic reward from visitation counts. Indeed, as the goal of an exploratory agent is to go to less-visited states, an intrinsic reward can be any decreasing function of the visitation counts. In the literature, the inverse of the square root of the visitation counts is known to be theoretically sound (Azar et al., 2017). Therefore, for a given time $t$, the intrinsic reward associated to the transition $(o_t, a_t, o_{t+1})$, can be defined as:

$$r_t = \left( \sqrt{N_\mathcal{K}(M, e_{t+1}, \{w_l\}_{l=1}^{|M|})} + c \right)^{-1}, \tag{2}$$

where a small constant $c \in \mathbb{R}_+$ is added to avoid numerical instability.

## 3 RECODE

In this section, we introduce our Robust Exploration via Clustering-based Online Density Estima-tion (RECODE) approach that computes intrinsic rewards for exploration. At a high level, RECODE stores a fixed number of weighted atoms (typically $5 \cdot 10^4$ or $2 \cdot 10^5$ depending on the domain) that are interpreted in the following as cluster-centers, along with their counts. The update rule of our memory $M$ for each new embedding $e \in \mathcal{E}$ and the computation of the intrinsic reward are detailed in Algorithm. 1. To each atom/cluster-center $f_l \in \mathcal{E}$, we associate a count $c_l \in \mathbb{N}$ (initialized to 0) that is updated after each new embedding $e$ is observed. When a new embedding is inserted we prob-abilistically choose either to add it as a new cluster (if it is far away from existing clusters relative to an adaptive threshold) or increment the count of its nearest neighbor and adjust the cluster center toward the new embedding. The update rule has a close connection to the DP-means algorithm of Kulis & Jordan (2011), with two key differences:

- the counts of the cluster-centers are discounted at each step, allowing our approach to deal with the non-stationarity of the data due both to changes in the policy and the embedding function, effectively reducing the weight of stale cluster-centers in the memory,
- when creating a new cluster-center, we remove an underpopulated one, so as to keep the size of the memory constant. We compare the qualitative behavior of different removal strategies in Fig. 8

A theoretical analysis is sketched in Appendix D. To help build some intuition about the quality of our density estimation, we illustrate in Figure 1 the result of Algorithm 1 on a toy example with

---

**Algorithm 1: RECODE**

**Input** : Embedding $e$, Memory $M = \{f_l\}_{l=1}^{|M|}$, cluster-center counts $\{c_l\}_{i=l}^{|M|}$, number of neighbors $k$, relative tolerance $\kappa$, squared distance estimate $d_m^2$, decay rate $\tau$, discount $\gamma$, insertion probability $\eta$, kernel function $\mathcal{K}$, intrinsic reward constant $c$

**Output:** Updated memory $M = \{f_l\}_{l=1}^{|M|}$, updated cluster-center counts $\{c_l\}_{i=l}^{|M|}$, updated squared distance $d_m^2$, intrinsic reward $r$

1 Compute weighted smoothed visitation-count of $e$: $\mathcal{N}_\mathcal{K}(M, e, \{1+c_l\}_{l=1}^{|M|}) = \sum_{l=1}^{|M|}(1+c_l)\,\mathcal{K}(f_l, e)$

2 Compute intrinsic reward $r = \left(\sqrt{\mathcal{N}_\mathcal{K}(M, e, \{1+c_l\}_{l=1}^{|M|})} + c\right)^{-1}$

3 Find nearest $k$ cluster centers to the embedding $e$: $N_k(e)$

4 Update squared distance estimate: $d_m^2 \leftarrow (1-\tau)\,d_m^2 + \frac{\tau}{k}\sum_{f \in N_k(e)}\|e - f\|_2^2$

5 Discount all cluster-center counts $c_l \leftarrow \gamma\,c_l \quad \forall l \in \{1, \cdots, |M|\}$

6 Find index of nearest cluster center $i = \arg\min_{l=1\cdots|M|}\|f_l - e\|_2$

7 Sample uniformly a real number in $[0,1]$: $u \sim U[0,1]$

8 **if** $\|f_i - e\|_2^2 > \kappa\,d_m^2$ and $u < \eta$ **then**

10      Sample index $j$ of cluster center to remove with probability $P(j) \propto 1/c_j^2$          // Remove under-populated cluster

11      Find index of nearest cluster center to $f_j$: $n = \arg\min_{l=1\cdots|M|,l\neq j}\|f_l - f_j\|_2$

12      Redistribute the count of removed cluster center: $c_n \leftarrow c_j + c_n$

13      Insert $e$ at index $j$ with count 1: $f_j \leftarrow e\,,\,c_j \leftarrow 1$          // Create a new cluster

15 **else**

16      Update nearest cluster center $f_i \leftarrow \frac{c_i}{c_i+1}f_i + \frac{1}{c_i+1}e$

17      Update nearest cluster-center count $c_i \leftarrow c_i + 1$

18 **end**

---

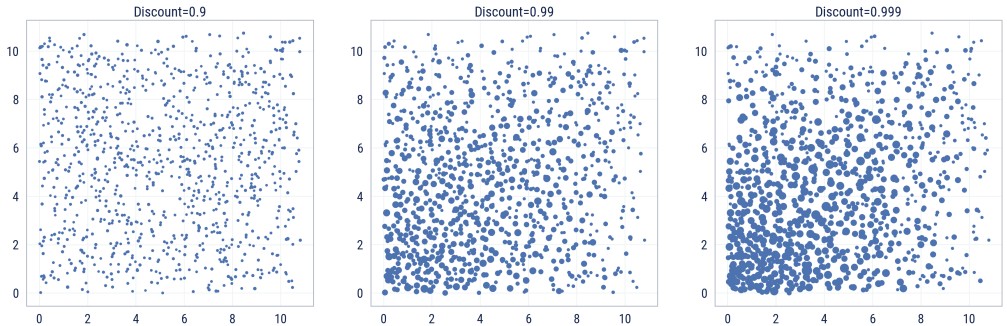

Figure 1: Density estimation using RECODE on a toy example: for step $t = 0, \ldots, 100$, we sample a batch of 64 2D-embeddings uniformly from the square of side $1+\sqrt{t}$. The support of the embedding distribution therefore expands over time to simulate a non-stationary distribution akin to the distribution of states visited by an RL agent over the course of exploration. We plot the clusters learned by RECODE with a size proportional to their count. We find that for a small enough discount, RECODE exhibits a short-term memory, accurately approximating the distribution of the final distribution. As we increase the discount, RECODE exhibits a longer-term memory, approximating the historical density of states, as can be seen by the concentration of probability mass in the bottom-left corner.

a non-stationary embedding distribution. We find in particular that tuning the discount allows to smoothly interpolate between short-term and long-term memory, a property that will prove crucial to achieve the strong experimental results of Section 5.

Note that contrary to the usual episodic memory used in Badia et al. (2020b;a), the memory is never reset, and is shared between all actors in our distributed RL agent. We use the following kernel function defined for $(e, f) \in \mathcal{E}^2$ by:

$$\mathcal{K}(f, e) = \frac{\epsilon}{\epsilon + \frac{\|e-f\|_2^2}{d_m^2}}\,\mathbb{1}_{\left\{\|e-f\|_2^2 < d_m^2\right\}}, \tag{3}$$

where $\epsilon \in \mathbb{R}_+$, $d_m^2$ is an estimate of the squared distance between an embedding and its nearest neighbors in the memory (see Algorithm. 1)) and $\mathbb{1}_{\{.\}}$ is the indicator function. Compared to Badia et al. (2020b;a), our smoothed visitation-counts computes the normalized distances from a given embedding $e$ to all its neighbors within a $d_m^2$-ball, instead of only the $k$-nearest neighbors. This change avoids an undesirable property of the $k$-NN approach where inserting a cluster with unit-

count close to the embedding $e$ might *reduce* its pseudo-count instead of increasing it if the $k$-th nearest neighbor of $e$ had a large count.

We now detail how `RECODE` can be integrated in a typical distributed RL agent (Espeholt et al., 2018; Kapturowski et al., 2018) that comprises several processes that run in parallel and interact with each other. Classically, a Learner performs gradient steps to train a policy $\pi_\theta$ and an embedding (representation) function $f_\theta$, forwarding the parameters $\theta$ to an Inference Worker. A collection of independent Actors query the inference worker for actions that they execute in the environment and send the resulting transitions to the Learner, optionally through a (prioritized) Replay (Mnih et al., 2015; Schaul et al., 2015). When using `RECODE`, the Actors additionally communicate with a shared Memory implementing Algorithm 1: at each step $t$, they query from the Inference Server an embedding $f_\theta(h_t)$ of their history and send it to the shared Memory which returns an intrinsic reward $r_t$ that is then added to the extrinsic reward to train the policy in the Learner process. In practice, we normalize the intrinsic reward by a running estimate of its standard-deviation as in Burda et al. (2019). A diagram giving an overview of a distributed agent using `RECODE` is given in Fig. 10.

## 4    REPRESENTATION LEARNING METHODS

The Learner process uses the data sent by the actors to train the policy $\pi_\theta$ via an RL algorithm and to train the embedder $f_\theta$ via a representation learning loss. As noted in Section 2, the choice of the embedding function $f_\theta : \mathcal{H} \to \mathcal{E}$ induces a metric in the embedding space $\mathcal{E}$ allowing to compare histories. Many different representation learning techniques have been studied in the context of exploration in RL (Burda et al., 2018a; Guo et al., 2020; 2022; 2021; Erraqabi et al., 2021). In the following, we focus on action prediction embeddings, introducing first the standard 1-step prediction formulation (Pathak et al., 2017; Badia et al., 2020b;a). Our embedding function $f_\theta$ is parameterized as a feed-forward neural network taking $o_t$, the observation at time $t$, as inputs. We further define a classifier $g_\phi$ that, given the embeddings of two consecutive observations $f_\theta(o_t), f_\theta(o_{t+1})$, outputs an estimate $p_{\theta,\phi}(a_t|o_t, o_{t+1}) = g_\phi\left(f_\theta(o_t), f_\theta(o_{t+1})\right)$ of the probability of taking an action given two consecutive observations $(o_t, o_{t+1})$. Both $f_\theta$ and $g_\phi$ are then jointly trained by minimizing an expectation of the following loss:

$$\min_{\theta,\phi} \mathcal{L}(\theta, \phi)(a_t) = -\ln(p_{\theta,\phi}(a_t|o_t, o_{t+1})),  \tag{4}$$

where $\mathcal{L}(\theta, \phi)(a_t)$ is the negative log likelihood and $a_t$ is the true action taken between $o_t$ and $o_{t+1}$. These embeddings have been shown to be helpful in environments with many uncontrollable features in the observation (Badia et al., 2020b), such as the game of *Pitfall!* in Atari, where they might result in spurious sources of novelty even when the agent is standing still.

We note however that `RECODE` can be used with an arbitrary embedding function, e.g. one tailored for the domain of interest. One downside of the standard, 1-step action-prediction method is that the simplicity of the prediction task may only require highly localized and low-level features to be learned for its solution, which may not be informative of more geometrical or topological notions of environment structure, that partially-observable or 3D-exploration tasks might require. Other popular forms of representation learning such as Contrastive Predictive Coding (CPC, Oord et al. (2018)) or Predictions of Bootstrapped Latents (PBL, Guo et al. (2020)) utilize temporally-extended prediction tasks but do not enforce any notion of controllability. We now present a novel generalization of action-prediction embeddings that we show to yield strong results on `DM-HARD-8` in section 5.

A straightforward generalization of 1-step action prediction is to predict sequences of actions between observations $o_t$ and $o_{t+k}$, but in general there may be many such sequences which are possible, besides the one which is obtained by the behavior policy. This introduces an additional policy-dependent non-stationarity in the prediction task which could potentially hinder learning efficiency and stability. To counteract this problem we could provide all but one of the intervening actions to our context $(o_t, a_t, a_{t+1} \ldots a_{t+k-2}, o_{t+k})$, and predict only $a_{t+k-1}$. However, in partially observed domains it is possible that these two observations alone are insufficient to accurately localize the agent's state, and therefore it may be beneficial to provide additional context before $o_t$.

Concretely, we propose to apply a causally-masked transformer to sequences of observation and action embeddings, such that at each timestep $t$ exactly one of $o_t$ and $a_t$ is provided. The transformer output is then projected down to the size of the embedding ($\dim z_t = \dim f_\theta(o_t)$), and the difference

between the two is input into a final MLP classifier, $g_\phi$. During training, we randomly sample $N = 4$ masks per trajectory to help reduce gradient variance. Note that we use $e_t = f_\theta(o_t)$, the transformer *inputs*, as the embeddings for RECODE in order to avoid leaking information about the agent's trajectory. As with 1-step action prediction, we train the representation using maximum likelihood. We refer to this approach as Coupled Action-State Masking (CASM) in the following. Figure 2 shows a diagram of the architecture.

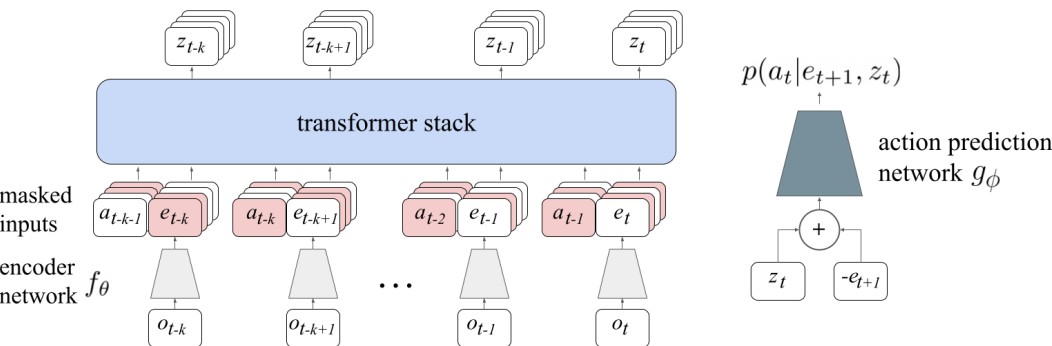

Figure 2: Coupled Action-State Masking (CASM) architecture used for learning representations in partially observable environments. Note that masked inputs are shaded in pink.

## 5 EXPERIMENTS

In this section, we experimentally validate the generality of our approach by applying it to several domains with distinct properties. We first show that we can match or improve state-of-the-art learning efficiency on the hardest exploration games contained in the well-established Atari benchmark. We then turn to DM-HARD-8, a suite of partially-observable 3D hard exploration tasks, where we obtain results that match or exceed the recently proposed BYOL-Explore (Guo et al., 2022) agent. All the candidates evaluated in the experiments in the main paper and the extended experiments in App. E are composed by three main modules: (1) a base agent, responsible for core RL aspects such as collecting observations and updating the policy, (2) an algorithm responsible to generate an exploration bonus, and (3) a representation learning algorithm responsible to generate latent representation of the observations that allow the exploration bonus to be meaningful. Each of the agent names in the experiments will reflect which specific algorithm is responsible for each of the three modules. For example, in our more detailed taxonomy the original MEME agent described in Kapturowski et al. (2022) is denoted as the MEME-NGU-AP baseline. Similarly, our MEME-RECODE-AP uses the same base agent (MEME) and representation learning algorithm (AP), with the only modification being the intrinsic reward mechanism being swapped from NGU to RECODE. For a full description of all the baselines we refer the reader to App. A. We use a memory comprised of $5 \cdot 10^4$ atoms for our Atari experiments and $2 \cdot 10^5$ atoms for our DM-HARD-8 experiments. We find that the resulting agent runs at roughly the same speed as the original MEME agent. More hyperparameters values can be found in App. A.

### 5.1 ATARI

The Atari Learning Environment (ALE, Bellemare et al., 2013) is one of the most used RL benchmarks for deep RL, comprising 57 Atari games which are mostly 2-D, and have a limited degree of partial observability which can often be rectified by stacking a small number of frames. On the other hand, many games have a long optimization horizon (with episodes lasting up to 27000 steps using the standard action-repeat of 4), and rewards vary considerably in both scale and density. Among the 57 Atari games, only a few are considered hard-exploration (Bellemare et al., 2016) such as Montezuma's Revenge, Pitfall and Private Eye. For evaluation, we follow the classical 30 random no-ops evaluation regime (Mnih et al., 2015; Van Hasselt et al., 2016), and average performance over 3 seeds. This evaluation regime does not use sticky actions (Machado et al., 2018).

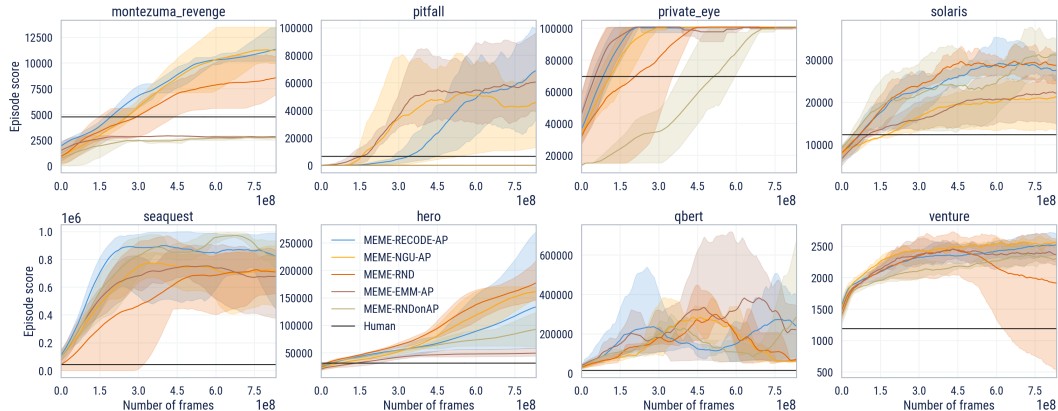

Figure 3: Comparison of `RECODE` against other exploration bonuses using a `MEME` base agent on 8 hard exploration games from the Atari domain.

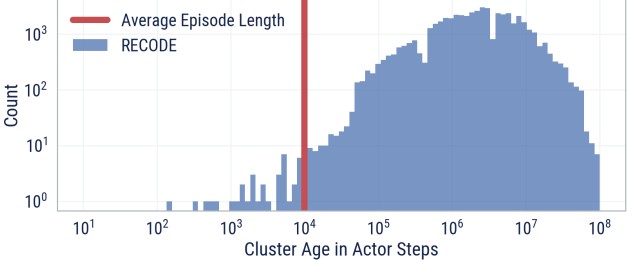

Figure 4: Age distribution of the clusters learned by `RECODE` on Montezuma's Revenge. We set $\gamma = 0.999$ as in the experiments of Figure 3. We indicate in red the average length of an episode, showing that in this setting, `RECODE`'s memory reaches back thousands of episodes.

We compare our approach with `MEME-NGU-AP` and its ablations on 8 of the hardest exploration games on Fig. 3. We find that `MEME-RECODE-AP` matches the performance of the original `MEME` agent with a single, simpler intrinsic reward, achieving super-human performance on all 8 hard exploration games. Indeed, as shown in Fig. 3, `MEME-NGU-AP` requires the full `NGU` exploration bonus (i.e., both `RND` and `EMM`) to solve all 8 games: `RND` on its own cannot solve Pitfall! because of the many uncontrollable features in its observations, while `EMM` on its own cannot solve Montezuma's Revenge because it requires long-term memory. Because `RECODE` estimates the visitation counts over many episodes using `AP` embeddings that discard uncontrollable dynamics, it is able to solve both games with a single intrinsic reward. We probe how far back the memory of `RECODE` goes in Montezuma's Revenge in Figure 4 and find that the distribution of the age of the clusters learned by `RECODE` exhibits a mode around $2 \cdot 10^6$ actor steps, which corresponds to hundreds of episodes, with a significant number of clusters ten times older than that. We also compare our approach with `MEME-RNDonAP` (detailed in Appendix E), a modification of `RND` built on top of `AP` embeddings, in an attempt to fix the aforementioned undesirable properties of `RND`. We find that this approach does not allow to solve some of the hardest games such as Montezuma's Revenge or Pitfall!. One possible explanation of this failure is the fact that a large `RND` error can be caused by either the observation of a new state, or a drift in the representation of an already observed one. The failure of `RND` to disentangle these two effects results in poor exploration.

## 5.2 NOISY MONTEZUMA'S REVENGE

Atari Games are deterministic with respect to the RAM state, which is a property that most real-world environments do not share (e.g.: observations may be noisy due to imperfect sensors). As `RND` relies on predicting a random embedding of the raw observation to determine whether a state is new or not, it cannot learn to discard noisy features, hindering its ability to detect meaningful novelty

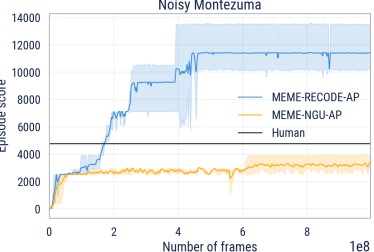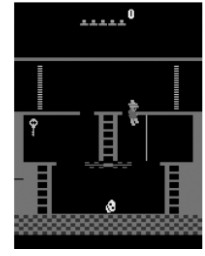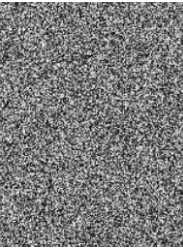

Figure 5: (Left) : Performance of `RECODE` compared to `NGU` on Noisy Montezuma. (Middle and Right) : A frame of Noisy Montezuma where the noise is concatenated to the original frame.

in the presence of noise. In this section, we show that `MEME-NGU-AP` inherits the limitations of `RND`. To this end, we consider the game of Montezuma's Revenge, for which effective exploration requires long-term memory as shown by the ablations of Section 5.1. We challenge `RND` by concatenating the original grey-scale, $210 \times 160$ pixels frame with a noisy frame of the same shape, where each pixel is sampled uniformly at random from the range $[0, 255]$. This type of noise is commonly referred to as noisy TV (Pathak et al., 2017). The results of our experiments on this environment are presented on Figure 5. Perhaps as expected, we find that the performance of `MEME-NGU-AP` is strongly deteriorated, since it relies on `RND` for the long-term component of its exploration bonus, and performs no better than a pure RL baseline without exploration bonus (Kapturowski et al., 2018). `RECODE`, on the other hand, relies on action-prediction embeddings to estimate global visitation-counts. In this embedding space, states that only differ in the noisy, uncontrollable part of the observation tend to be aliased together, so that the effect of the noise on exploration vanishes. Indeed, the performance of `MEME-RECODE-AP` on Montezuma's Revenge is unchanged when adding noise.

### 5.3  `DM-HARD-8`

`DM-HARD-8` (Gulcehre et al., 2019) is a benchmark comprised of 8 hard exploration tasks, originally built to emphasize the difficulties encountered by an RL agent when learning from sparse rewards in a procedurally-generated 3-D world with partial observability, continuous control, and highly variable initial conditions. Each task requires the agent to interact with specific objects in its environment in order to reach a large apple that provides reward (see Figure 11 in the Appendix for an example). Being procedurally-generated, properties such as object shapes, colors, and positions are different every episode. A recently proposed exploration bonus called `BYOL-Explore` (Guo et al., 2022) was shown to be effective on this domain, while previous successes were only achieved through the use of human demonstrations (Gulcehre et al., 2019).

We first assess the effect of using the `RECODE` bonus over `NGU` by performing a drop-in replacement of the intrinsic reward in the `MEME` base agent as in Section 5. We consider the performance of the resulting agent in the single-task version of `DM-HARD-8` since the `MEME` base agent is not designed to work in the multi-task setting out-of-the-box. The results, presented on Fig. 6, show that `MEME-RECODE-AP` approach can solve 4 out of 8 games, strictly improving over the original `MEME` agent using either `NGU` or `EMM` in this direct comparison. Second, we turn to using the `CASM` representation learning approach introduced in Section 4 for our embedding. We find that the resulting `MEME-RECODE-CASM` agent attains state-of-the-art results on the `DM-HARD-8` benchmark, reliably solving 6 out of 8 games. This matches and often improves over the state-of-the-art performance of Guo et al. (2022), which combined the `BYOL-Explore` exploration bonus with a `VMPO`-based base agent. Finally, to demonstrate the generality of our approach, we also test the combination of a `VMPO` base agent similar to that used in Guo et al. (2022) with `RECODE` and `AP` embeddings. The resulting agent is evaluated in the multi-task setting, reliably solving 4 games with zero additional tuning (App. G), matching the result obtained with Human Demonstrations in Gulcehre et al. (2019).

## 6  CONCLUSION AND DISCUSSION

We have introduced Robust Exploration via Clustering-based Online Density Estimation (`RECODE`), a principled yet simple exploration bonus for deep Reinforcement Learning (RL) agents that allows

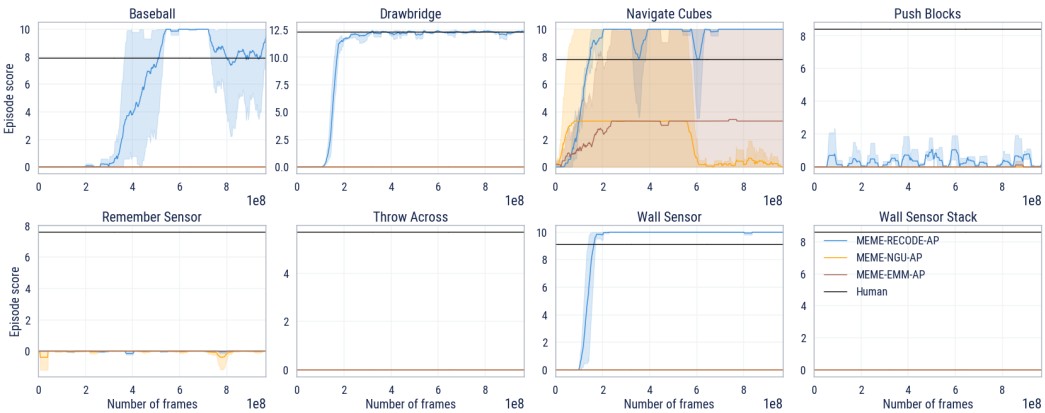

Figure 6: Performance of `RECODE` compared to `MEME` on the single-task version of `DM-HARD-8`.

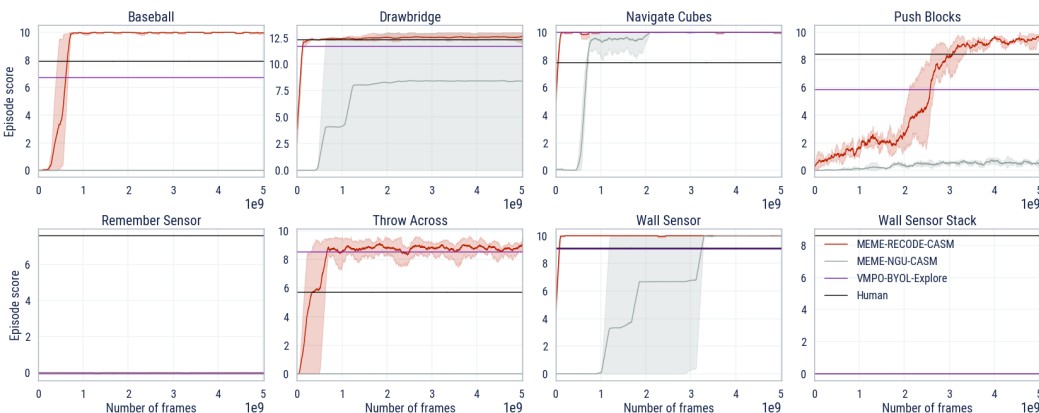

Figure 7: Performance of `RECODE` using `CASM` embeddings compared to BYOL-Explore on the single-task version of `DM-HARD-8`. The BYOL-Explore results correspond to the final performance reported in Guo et al. (2022), after $10^9$ environment frames, averaged over 3 seeds.

to perform robust exploration by computing visitation counts from a slot-based memory. Contrary to previous work, where the memory was short-term (i.e. only able to attend to the current episode due to memory limit), our memory is able to model a wide range of timescales determined by the choice of discount. This is made possible by the use of an online clustering algorithm that is able to approximate the density of visited states and from which we derive a simple and easy-to-tune intrinsic reward.

We evaluate our exploration bonus on top of the recent `MEME` agent, which is a value-based agent making use of the `NGU` intrinsic reward, that achieves state-of-the-art results on the Atari domain. We show that we can replace the complex `NGU` intrinsic reward that combines `RND` and episodic memory with the simpler `RECODE` reward without loss of performance on the hardest exploration levels in Atari. Furthermore, we highlight important failure modes of `NGU` in the presence of noise, and show that `RECODE`'s performance is unaffected. Similarly, we highlight the limitations of `NGU` in procedurally-generated 3D environments such as `DM-HARD-8`, and demonstrate the improvements brought by using `RECODE` as an intrinsic reward. Next we introduce a novel representation learning method better suited to 3D and partially observable domains, which provides a significant boost in performance on this task suite, enabling to achieve a new state-of-the-art on `DM-HARD-8` in the single-task setting.

Importantly we note that `RECODE` is agnostic to the choice of embeddings $f_\theta(h_t)$, and while `CASM` presents one compelling option to extend 1-step action prediction, we hypothesize that much progress could be made in challenging exploration problems by developing more sophisticated representations which can be used in conjunction with `RECODE`.

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

## A HYPER-PARAMETERS

We omit hypers which do not differ from the base agent, MEME.

Table 1: Atari Hyper-parameters.

| Parameter | Value |
|---|---|
| RECODE memory size | $5 \times 10^4$ |
| RECODE discount $\gamma$ | 0.999 |
| RECODE relative tolerance $\kappa$ | 0.01 |
| RECODE insertion probability $\eta$ | 0.05 |
| RECODE reward constant $c$ | 0.1 |
| RECODE decay rate $\tau$ | 0.9999 |
| RECODE neighbors $k$ | 20 |
| IM Reward Scale $\beta_{\text{IM}}$ | 1.0 |
| Max Discount | 0.9997 |
| Min Discount | 0.97 |
| Replay Period | 80 |
| Trace Length | 160 |
| Replay Ratio | 6.0 |
| Replay Capacity | $2 \times 10^5$ trajectories |
| Batch Size | 64 |
| RL Adam Learning Rate | $3 \times 10^{-4}$ |
| Emedding Adam Learning Rate | $6 \times 10^{-4}$ |
| RL Weight Decay | 0.05 |
| Embedding Weight Decay | 0.05 |
| RL Torso initial stride | 4 |
| RL Torso num blocks | $(2, 3, 4, 4)$ |
| RL Torso num channels | $(64, 128, 128, 64)$ |
| RL Torso strides | $(1, 2, 2, 2)$ |

Table 2: DM-HARD-8 Hyper-parameters.

| Parameter | Value |
|---|---|
| RECODE memory size | $2 \times 10^5$ |
| RECODE discount $\gamma$ | 0.997 |
| RECODE relative tolerance $\kappa$ | 0.2 |
| RECODE insertion probability $\eta$ | 0.2 |
| RECODE reward constant $c$ | 0.01 |
| RECODE decay rate $\tau$ | 0.9999 |
| RECODE neighbors $k$ | 20 |
| IM Reward Scale $\beta_{\text{IM}}$ | 0.1 |
| Max Discount | 0.997 |
| Min Discount | 0.97 |
| Replay Period | 40 |
| Trace Length | 80 |
| Replay Ratio | 2.0 |
| Replay Capacity | 5000 trajectories |
| Batch Size | 128 |
| RL Adam Learning Rate | $1 \times 10^{-4}$ |
| Embedding Adam Learning Rate | $3 \times 10^{-4}$ |
| RL Weight Decay | 0.1 |
| Embedding Weight Decay | 0.1 |
| RL Torso initial stride | 2 |
| RL Torso num blocks | $(2, 4, 12, 6)$ |
| RL Torso num channels | $(64, 128, 128, 64)$ |
| RL Torso strides | $(1, 2, 2, 2)$ |

Table 3: `CASM` Hyper-parameters.

| Parameter | Value |
|---|---|
| Transformer Type | GatedTransformerXL |
| State Mask Rate | 0.8 |
| Num Masks Per Trajectory | 4 |
| Action Embedding Size | 32 |
| Num Layers | 2 |
| Attention Size | 128 |
| Num Attention Heads | 4 |
| MLP Hidden Sizes | (512,) |
| Predictor Hidden Sizes | (128,) |

## B  AGENT TAXONOMY

All methods evaluated in the experiments in the main paper and the extended experiments in App. E are composed by three main components.

- A base agent that oversees the overall RL learning process (e.g., executing actions and collection observations, computing adjusted returns, updating the policy, ...). We focus on `MEME` (Kapturowski et al., 2022), a recent improvement over Agent57 (Badia et al., 2020a) that achieves much greater sample efficiency and is the current state-of-the-art on Atari, and a VMPO-based agent (Guo et al., 2022) that is the current state-of-the-art on `DM-HARD-8`.

- An algorithm to generate intrinsic rewards. In addition to `RECODE`, we also consider the recent `BYOL-Explore` Guo et al. (2022), `NGU` Badia et al. (2020b) and `NGU`'s two building blocks, `RND` Burda et al. (2018b) and Episodic Memory (`EMM`) Pritzel et al. (2017).

- A representation learning mechanism to generate observation embeddings which are fed to the intrinsic reward generator. We consider both Action Prediction (`AP`) and `CASM` embeddings. Note that some intrinsic reward modules cannot make effective use of the representation learning module (e.g., `RND`), while others merge both second and third module in a single approach (e.g., `BYOL-Explore`)

For example, in our more detailed taxonomy the original `MEME` agent described in Kapturowski et al. (2022) is denoted as the `MEME-NGU-AP` baseline, and compared against our novel `MEME-RECODE-AP` agent where the only modifications is the changed exploration reward. Table E reports more details on all combinations available present in our experiments.

Table 4: Taxonomy of agents used in the experiments.

| Agent name | | Base agent | Intrinsic reward | Representation learning |
|---|---|---|---|---|
| MEME-NGU-AP | Kapturowski et al. (2022) | MEME | NGU | AP |
| MEME-RND | (ablation) | MEME | RND | N/A[a] |
| MEME-EMM-AP | (ablation) | MEME | EMM | AP |
| MEME-RNDonAP | (ablation) | MEME | RND | AP [b] |
| MEME-RECODE-AP | (this paper) | MEME | RECODE | AP |
| MEME-RECODE-CASM | (this paper) | MEME | RECODE | CASM |
| MEME-NGU-CASM | (ablation) | MEME | NGU | CASM |
| VMPO-BYOL-Explore | Guo et al. (2022) | VMPO | BYOL-Explore | BYOL-Explore [c] |

(a) As in the original paper `RND` takes as input raw observations.

(b) To test `RND`'s ability to cope with non-stationary representations, we train an `AP` encoder concurrently with the policy and use it to create embeddings of the observations that are fed in `RND` (i.e., running `RND` on top of `AP`).

(c) The `BYOL-Explore` mechanism internally trains a neural network to predict the dynamical evolution of the observations. This provides the agent with both a reward/novelty signal (prediction error) as well as an embedded representation of the observations (that can be extracted from the last few layers of the network).

## C    RELATED WORKS

In this section, we give a brief and non-exhaustive overview of past works computing visitation counts or estimating densities in RL. We classify them as either parametric or non-parametric.

**Parametric methods.**    Bellemare et al. (2016) and Ostrovski et al. (2017) propose to compute pseudo-visitation counts using density estimators on images such as Context Tree Switching (CTS,Bellemare et al., 2014) or PixelCNN (Van den Oord et al., 2016). On the other hand, Tang et al. (2017) use locality-sensitive hashing to map continuous states to discrete embeddings, where explicit visitation counts are computed. Some methods such as RND (Burda et al., 2019) can be interpreted as estimating implicitly the density of observations by training a neural network to predict the output of a randomly initialized and untrained neural network which operates on the observations. Hazan et al. (2019); Pong et al. (2019); Lee et al. (2019); Guo et al. (2021) propose algorithms that search a policy maximizing the entropy of its induced state-space distribution. In particular, the loss optimized by Guo et al. (2021) allows to compute a density estimate as well as maximizing the entropy. Finally, Domingues et al. (2021b) computes a density estimation on top of learned representations, which are inspired by bonuses used in reward-free finite MDPs.

**Non-parametric methods.**    Non-parametric density estimates that we build on date back to Rosenblatt (1956); Parzen (1962) (Parzen–Rosenblatt window) and are widely used in machine learning as they place very mild assumptions on the data distribution. Non-parametric, kernel-based approaches have been already used in RL and shown to be empirically successful on smaller scale environments by Kveton & Theocharous (2012) and Barreto et al. (2016) and are theoretically analyzed by Ormoneit & Sen (2002); Pazis & Parr (2013); Domingues et al. (2021a). In NGU (Badia et al., 2020b), Agent57 (Badia et al., 2020a) and MEME (Kapturowski et al., 2022), a non-parametric approach is used to compute a short term reward at the episodic level. Liu & Abbeel (2021) propose an unsupervised pre-training method for reinforcement learning which explores the environment by maximizing a non-parametric entropy computed in an abstract representation space. The authors show improved performance on transfer in Atari games and continous control tasks. Seo et al. (2021) use random embeddings and a non-parametric approach to estimate the state-visitation entropy, but do not generalize to concurrently learned embeddings. Tao et al. (2020) show that K-NN based exploration can improve exploration and data efficiency in model-based RL. While non-parametric methods are good models for complex data, they come with the challenge of storing and computing densities on the entire data set. We tackle this challenge in Sec. 3 by proposing a method that estimates visitation counts over a long history of states, allowing our approach to scale to much larger problems than those considered in previous works, and without placing assumptions on the representation, that can be trained concurrently with the exploration process and doesn't need to be fixed a priori.

## D    A CLUSTERING VIEW OF RECODE

The update rules of RECODE for its memory structure in Algorithm 1 can be interpreted as an approximate inference scheme in a latent probabilistic clustering model. We explore this connection here as means to better understand and justify the proposed algorithm as a density estimator. Briefly, our scheme is related to DP-means (Kulis & Jordan, 2011), with adaptations made to accommodate the additional complexities of our setting, which follows a streaming protocol (i.e., data must be explicitly consumed or stored as it arrives and data that are not stored cannot be accessed again) and is non-stationary (i.e., data are not assumed to be identically distributed as time advances). The clustering algorithm resulting from these adaptations is shown in Algorithm 2.

We first address the modifications introduced to deal with the memory limitations of the streaming setting: 1) each datum (embedding $e_t$ in our notation) is incorporated into a cluster distribution approximation, once, then discarded, 2) the total number of clusters is stochastically projected down onto an upper limit on the number of clusters (otherwise they would grow without bound–albeit progressively more slowly). Both modifications allow our method to maintain constant space complexity in the face of an infinite stream of data.

---

**Algorithm 2:** A streaming clustering algorithm.

---

1 **Parameters**
2 *Number of clusters $L$*
3 *Number of nearest cluster centres $k$*
4 *Discounting of counts at each step $\gamma$*
5 *Scaling of threshold on new cluster creation $\kappa$*
6 *Probability of actually creating new cluster when threshold exceeded $\eta$*
7 **State**
8 *Threshold on creating new cluster $\delta = 0$*
9 *Cluster centres $\mu_l = 0 \quad \forall l \in 1 \ldots L$*
10 *Cluster counts $c_l = 0 \quad \forall l \in 1 \ldots L$*
11 *Indices of $k$-nearest neighbours of point $e$: $N_k(e)$*
12 **Implementation**
13 **for each** received embedding $e \in \{e_0, e_1, e_2, \ldots\}$ **do**
14 $\quad$ Update threshold on inter-cluster distances $\delta \leftarrow (1 - \tau)\delta + \frac{\tau}{k}\sum_{l \in N_k(e)} \|\mu_l - e\|_2^2$
15 $\quad$ Discount all cluster-center counts $c_l \leftarrow \gamma\, c_l \quad \forall l \in 1, \ldots, L$
16
17 $\quad$ Find index of nearest cluster center $i = \arg\min_{l=1 \ldots L} \|\mu_l - e\|_2$
18 $\quad$ **if** $\|\mu_i - e\|_2^2 > \kappa\,\delta$ and with probability $\eta$ **then**
19 $\quad\quad$ Sample index $j$ of cluster center to remove with probability $P(j) \propto 1/c_j^2$
20 $\quad\quad$ Find index of nearest cluster center to $\mu_j$: $k = \arg\min_{l=1 \ldots L, l \neq j} \|\mu_l - \mu_j\|_2$
21 $\quad\quad$ Redistribute the count of removed cluster center: $c_k \leftarrow c_j + c_k$
22 $\quad\quad$ Replace cluster $j$ with a singleton of $e$: $\mu_j \leftarrow e\,, c_j \leftarrow 1$
23 $\quad$ **else**
24 $\quad\quad$ Update nearest cluster center $\mu_i \leftarrow \frac{c_i}{c_i+1}\mu_i + \frac{1}{c_i+1}e$
25 $\quad\quad$ Update nearest cluster-center count $c_i \leftarrow c_i + 1$
26 $\quad$ **end**
27 **end**

---

The *step-wise* justification of the Algorithm 2 is relative straightforward. At step $t$, for embedding $e_t$, we show that the following objective is minimised:

$$\min_{l \in 1, \ldots, L} \|\mu_l - e_t\|_2^2 \tag{5}$$
$$\text{s.t.} \quad \|\mu_l - e_t\|_2^2 \leq \kappa\delta$$

Working backwards: Updating the cluster center of the closest cluster reduces the objective directly and will not violate the constraint (unless it was already in violation; this excluded in the precondition of this branch). This accounts for the "else" branch. The "if" branch introduces a new cluster center precisely at $e_t$, thus equation 5 is minimised completely: it is zero for this branch. Finally, selecting the index of the nearest cluster center directly minimises placement of the branch according to equation 5, ignoring the constraint (which is latest ensured by the "if/else"). Note that the hard constraint of equation 5 takes the place of the soft cluster penalty of DP-means (Kulis & Jordan, 2011).

The updates to the cluster centers, unlike k-means and DP-means, are done in an exponentially-weighted moving average of the embeddings, rather than as global optimisation step utilising all of the data. Consequently, and importantly, what happens to equation 5 evaluated for $e_s$ where $s \neq t$, is of significant interest, as objectives for k-means and DP-means account for all data, rather than a single datum.

We now turn to this question, which will be answered by examining the modifications we introduced to deal with the non-stationarity of our data stream: 1) the cluster count decays, 2) merging two clusters to accommodate a new one, 3) the exponentially weighted moving average update of cluster centers.

In k-means, all of the data are retained. This makes k-means costly: at each step of fitting the entire data set is examined to update the cluster assignments and update the cluster means. Instead, we take a distributional approximation to the data associated with each cluster, and when re-adjusting cluster assignments according to equation 5, we do so in terms of this distributional approximation.

In particular, each cluster is approximated by a Gaussian distribution with precision 1 and whose mean is unknown but with prior zero and precision 1. Specifically:

$$m_l \sim \mathcal{N}(0, c_0)$$
$$e_i | m_l \sim \mathcal{N}(m_l, 1)$$

where $\mathcal{N}(\mu, \tau)$ denotes a Gaussian (or normal) distribution with mean $\mu$ and precision $\tau$ (precision is the inverse variance). Since the prior on $m_l$ is conjugate to the likelihood on $e_i$, we know that the posterior on $m_l$ will have the form $\mathcal{N}(\mu_l, c_l)$. Updating this posterior with a single embedding $e_i$ has the form:

$$\mu \leftarrow \frac{c_l}{c_l + 1}\mu + \frac{1}{c_l + 1}e_i$$
$$c_l \leftarrow c_l + 1$$

This is precisely the update in Algorithm 2.

Note that in this model, the counts $c_l$ are also the precision parameters of the distribution representing the inverse spread (or the concentration) of each cluster. At each step of Algorithm 2 these counts are decayed. Effectively, this causes the variance of the distribution representing each cluster to spread out: thus at each time step, each cluster becomes less concentrated and more uncertain about which data points belong to it. The hyperparameter $\gamma$ captures the rate of diffusion of all clusters in this manner. This uncertainty increase applied at each step acts as a "forgetting" mechanism that helps the algorithm to deal with a changing data distribution.

Cluster re-sampling, as already justified for $e_t$ above in terms of equation 5, ensures that the number of clusters is bounded by $L$. There are two details to examine: what is merged, and how it is merged. As $c_j \mapsto 0$, the probability assigned by the Gaussian likelihood of cluster $j$ to any new datum approaches zero also, thus the cluster with the lowest counts is likely to have the least impact on future density estimates (as it is most diffuse). When $c_j \gg 0$, however, it is not so clear which cluster should be removed, therefore, we stochastically select which cluster to remove inversely proportional to the square of the counts (using the square of the counts emphasizes small differences in counts more than $1/c_j$). The cluster could potentially be removed completely, but we instead choose to re-assign its counts to the nearest cluster as we experimentally found this strategy to be less sensitive to the choice of hyperparameters.

## E    RND ON TOP OF ACTION PREDICTION EMBEDDINGS

We adapt `RND` to leverage trained action-prediction embeddings, which we refer to as `RNDonAP`. To that effect, we use a randomly initialize Multi-Layer Perceptron (MLP) to perform a random projection of the embedding, and use a second, trained MLP, to reconstruct this random projection. The reconstruction error provides an intrinsic reward for exploration, which we normalize by a running estimate if its standard deviation as in Burda et al. (2019). We find that the resulting agent is unable to solve some of the hardest exploration games such as `Montezuma's Revenge` or `Pitfall`. The results of this ablation is shown in Fig. 9. Preliminary experiments with pre-trained embeddings do seem to indicate that `RNDonAP` can obtain stronger performance in this setting, but the inability to concurrently train the embeddings greatly limits the general applicability of the method.

## F    ARCHITECTURE OF AN AGENT USING RECODE

Figure 10 shows the typical architecture of a distribued RL agent using `RECODE`.

## G    MULTITASK EXPERIMENTS

We also implemented `RECODE` in a VMPO-based agent similar to the one used with BYOL-Explore (Guo et al., 2022), and compared our performance with BYOL-Explore in the multi-task

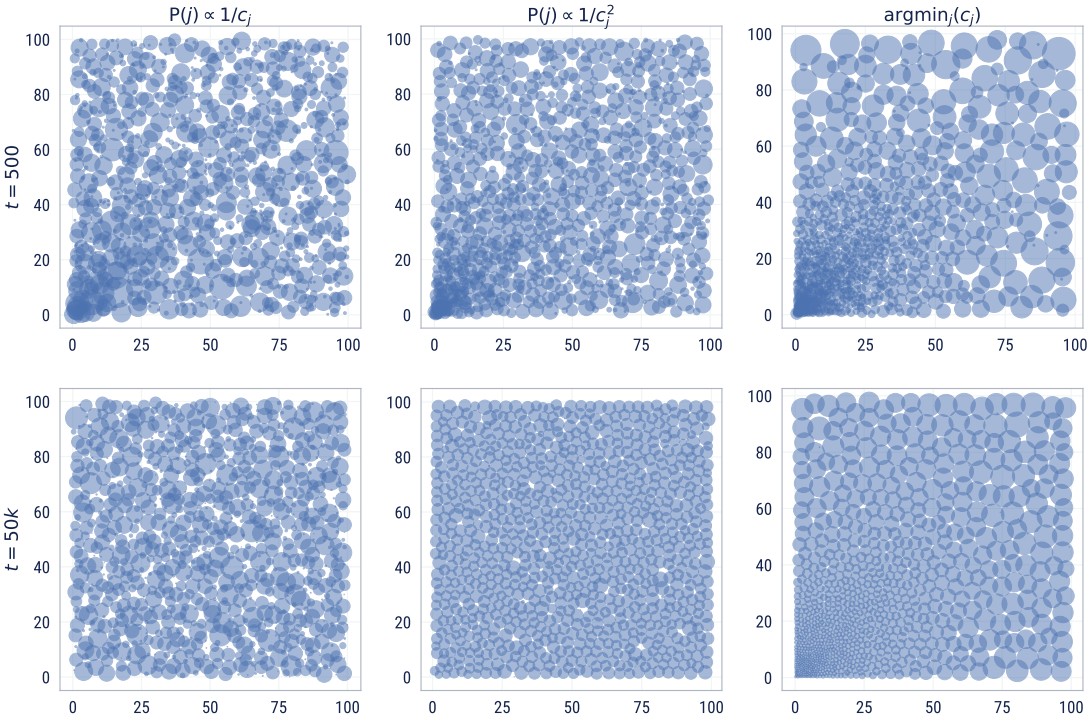

Figure 8: Effect of removal strategy on evolution of cluster centers and counts (with counts corresponding to the size of the marker). At each timestep $t$ we sample a batch of 64 2D-embeddings from a square of side $\min(100, t)$. After $t = 100$ the distribution remains stationary and we would like the distribution of cluster centers and counts to be to become approximately uniform after enough time has passed. For a deterministic removal strategy which selects the clusters with the lowest counts, the cluster centers can remain skewed long after the distribution has stopped changing. For both probabilistic removal strategies, the cluster centers become approximately uniform, but only for the $1/c_j^2$ removal strategy do we observe both cluster centers and counts become uniform. (Note that we use a discount of $0.9999$)

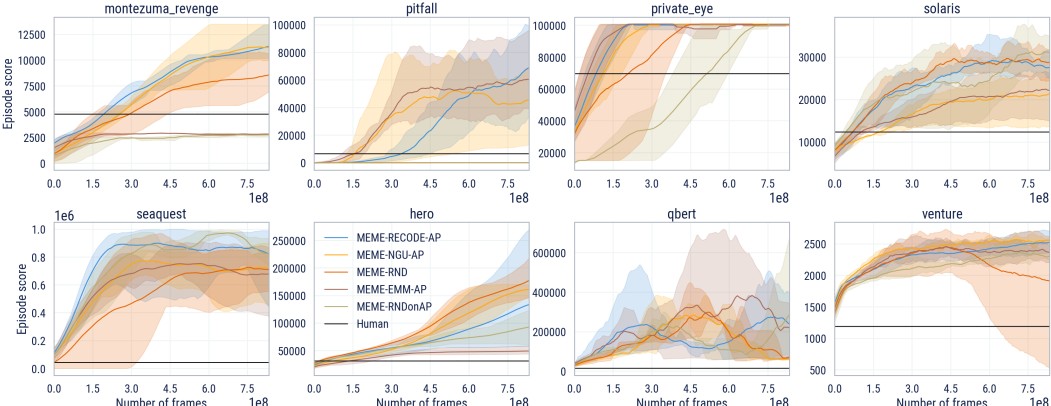

Figure 9: Performance of `RECODE` compared to `MEME` and its ablations on 8 hard exploration Atari games.

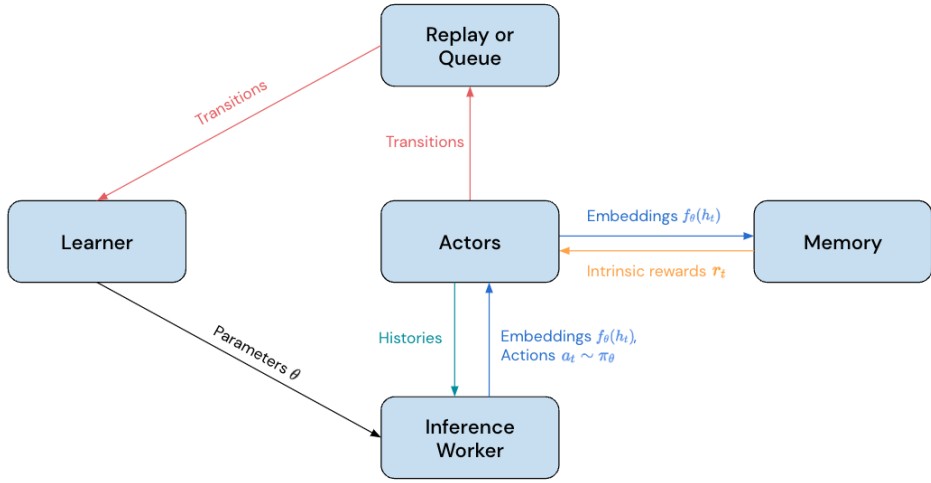

Figure 10: Overview of the architecture of a distributed agent using RECODE.

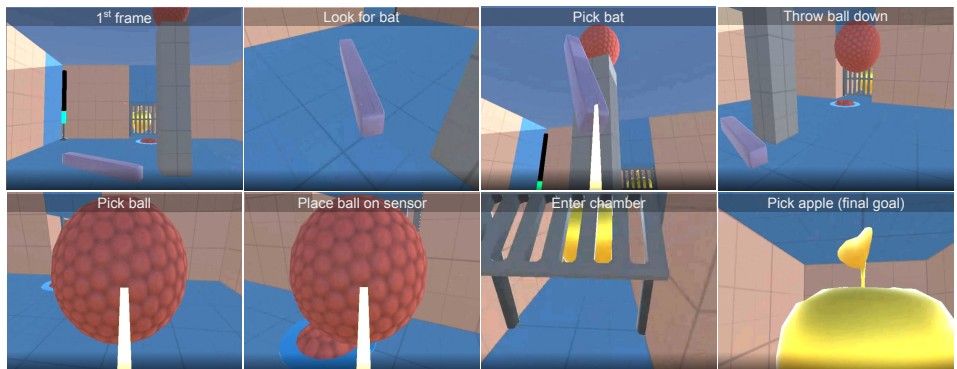

Figure 11: 1st-person-view snapshots of an agent solving the Baseball task. Images are ordered chronologically from left to right and top to bottom. Each image depicts a specific stage of the task.

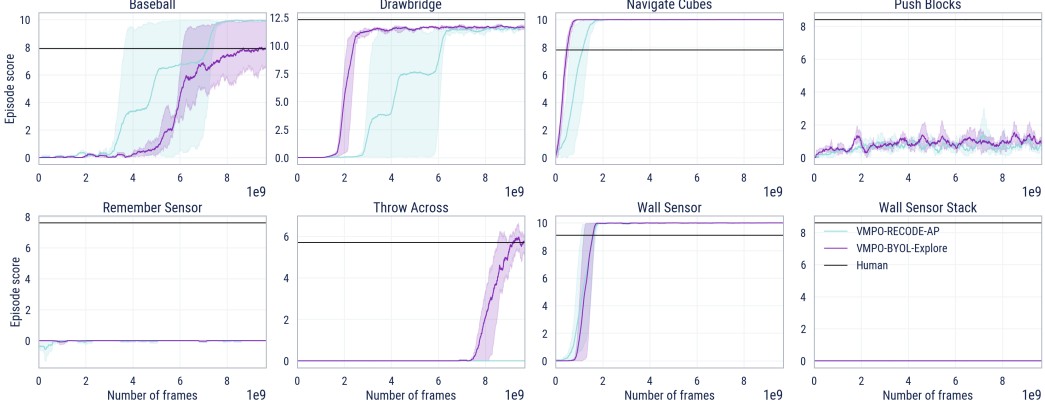

Figure 12: Performance of RECODE compared to BYOL-Explore on the multi-task version of DM-HARD-8. Our RECODE implementation in this experiments is based on VMPO, using a continuous action set.

setting. This experiment serves two different purposes. First, this demonstrates the generality of our exploration bonus, that is thus shown to be useful in widely different RL agents, be they value-based or policy-based. Second, we can do a direct comparison with the state of the art BYOL-Explore agent in the multi-task settings. We however note that the representation learning technique used in this experiment, 1-step Action Prediction, is based on a feed-forward embedding that discards past history, and may therefore not be the best fit for exploration in Partially Observable MDPs (POMDPs). Still, we show in Fig. 12 that the performance of RECODE is competitive with that of BYOL-Explore, with only one level missing to match its performance. Improving this performance using better-suited representations, such as CASM, is left for future work.

## H  AGGREGATED RESULTS

Here we present our main results aggregated over all environments in each task suite. To ensure that no single environment dominates due to larger reward scales we use the Human Normalized Score (Mnih et al., 2015) in each environment, and then cap scores above 100% prior to averaging. As can be observed in Fig. 13 (Left), the uncapped score can swing significantly over time, which in this case is simply an artifact the high variance present in *Q*bert*. [3]

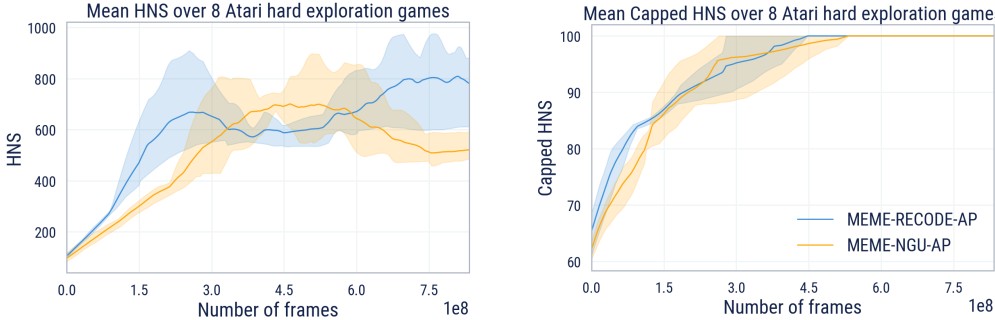

Figure 13: (Left) : Mean Human-Normalized Scores of MEME-RECODE-AP compared to MEME-NGU-AP on Atari games. (Right) : Capped Human-Normalized Scores.

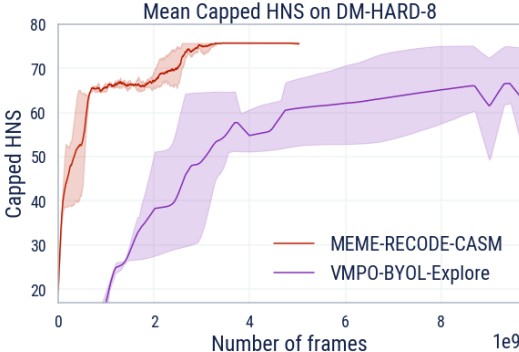

Figure 14: Mean Capped Human-Normalized Scores of MEME-RECODE-CASM compared to VMPO-BYOL-Explore on DM-HARD-8 games.

---

[3]This variance arises due to a bug in *Q*bert*, which allows for much larger scores to be obtained if exploited

