# OpenReview forum: "Robust Exploration via Clustering-based Online Density Estimation"
_ICLR.cc/2023/Conference — Submitted to ICLR 2023_

### Official Review · Reviewer_vYnp · 2022-10-24

**Confidence:** 4
**Correctness:** 2
**Technical Novelty And Significance:** 3
**Empirical Novelty And Significance:** 2
**Recommendation:** 3

**Clarity, Quality, Novelty And Reproducibility:**

The authors need to better clarify the distinctions between the contributions. They note "The first contribution of this paper is a general solution to (ii)." This is confusing. So does the "embedding function on observations or trajectories that encodes a meaningful notion of similarity"  include and estimate of smoothed visitation counts?

Highly non-standard notion of "Interaction Process between an Agent and its Environment". As defined, the history is infinite. Most work presumes a Markov property to bound the history. This raises an instant computational problem. Is it not critical to bound the history for computational or practical reasons?

The authors later imply that infinite histories are uninformative:
"When the embedding space is complex and too large, the vanilla visitation count becomes uninformative because each embedding is potentially different from all the atoms. To overcome this problem, soft-visitation counts are computed."

However, later on, the authors fix the history: "At a high level, RECODE stores a fixed number of weighted atoms (typically 5 · 104 or 2 · 105 depending on the domain) that are interpreted in the following as cluster-centers, along with their counts."

Further confusion is introduced in that the authors train a classifier g that, given the embeddings of *two* consecutive observations f(o_t), f(o_t+1), outputs an estimate p(a_t|o_t, o_t+1) = g? (f(o_t), f(o_t+1)).

Given this, it is difficult to understand the formal basis for this approach and its relationship to what is implemented, and why decisions are made.



It is confusing to introduce distributed learning "Note that contrary to the usual episodic memory used in Badia et al. (2020b;a), the memory is never reset, and it is shared between all actors when using a distributed RL agent."
Please just stick to explaining things without adding distracting details.

**Strength And Weaknesses:**

Strengths:
This is a challenging area, and the paper has clear novelty.


Weaknesses:

Overall, the ideas are interesting but the paper is poorly written and has inadequate experiments. The experiments should include a clear comparison with the standard Markov assumptions of RL and the general methods included here.

Theoretically, the paper is muddled and confusing. The authors need to distinguish the standard RL assumption of Markov models, vs. the exploration length allowed. This is unclear in the paper. The authors attempt to define "history" to incorporate exploration length allowed, but it is difficult to understand if this is actually the case.





**Summary Of The Paper:**

This paper focuses on  generalizing the action/prediction representation that leverages multi-step predictions and that we find to be better suited to a suite of challenging 3D-exploration tasks in DM-HARD-8. We show experimentally that our approach can work with a variety
of RL agents, and obtain state-of-the-art performance on Atari and DM-HARD-8.

The paper extends previous work,  limited to short-term memory (current episode), to address a wide range of timescales determined by the choice of discount. The authors propose an online clustering algorithm that is able to approximate the density of visited states.

**Summary Of The Review:**

The ideas are interesting and have novelty, but the paper is poorly written and has inadequate experiments. The experiments should include a clear comparison with the standard Markov assumptions of RL and the general methods included here.

---

> ### Author Response · Authors · 2022-11-18
> **Reply to reviewer concerns [Part 3]**
>
> > The authors later imply that infinite histories are uninformative: "When the embedding space is complex [...]"
>
> With this passage we do not mean to claim that infinite histories are uninformative. If the state space is large compared to the history, the probability of encountering the same state twice becomes vanishingly small. In the paragraph quoted here we try to communicate that when the size of the state-space grows much larger than the available history, some form of aggregation becomes necessary to recover meaningful counts.
>
> That is, the passage highlights that problems arise exactly when the space becomes too large w.r.t. the length of the history (i.e. amount of data collected), or in other words when histories are too short, not infinite.
>
> > However, later on, the authors fix the history: "At a high level, RECODE [...]"
>
> The history is never fixed (neither in length nor content), as it is only a notation. That passage refers to the memory mechanism in RECODE. That is, the information contained in a potentially infinite history has to be compressed and summarized (for the obvious computational reasons mentioned by the reviewer) in a finite memory. RECODE is an approach to do just that while trying to preserve as much information and performance in downstream RL tasks as possible.
>
> > Further confusion is introduced in that the authors train a classifier g that, given the embeddings of two consecutive observations f(o_t), f(o_t+1), outputs an estimate p(a_t|o_t, o_t+1) = g? (f(o_t), f(o_t+1)).
>
> Our domains consider discrete actions. Therefore the probability p(a_t|o_t, o_t+1) can be earned based on a classification problem (i.e. learning to classify which of the discrete actions was taken between two observations). In particular we consider a parametric class of functions (of which g is a member) that take as input two consecutive embeddings to output an estimate of each class probability (remember that f is the embedding function). In other words, the function g is how we model our posterior distribution over discrete actions.
> Following in the steps of Badia et al. we chose this formulation to train the embedding function f to encode the notion of controllability in the embeddings. In particular if a feature changes between f(o_t) and f(o_t+1) regardless of the action taken in between, it will not be useful for the prediction task. Therefore when f is optimized (through g) to predict actions it will naturally learn to encode features that correlate with aspects of the environment that we can impact through our actions (e.g. position of the agent on the screen) but not with aspects that we cannot control (e.g. a clock running in the corner or a far away background scene). This greatly reduces the implicit state-space and helps the agent to learn.
>
> > Given this, it is difficult to understand the formal basis for this approach and its relationship to what is implemented, and why decisions are made.
>
> We hope that our explanation of the notation will help the reviewer understand our choices.
>
> > It is confusing to introduce distributed learning "Note that contrary to the usual episodic memory used in Badia et al. (2020b;a), the memory is never reset, and it is shared between all actors when using a distributed RL agent." Please just stick to explaining things without adding distracting details.
>
> This is actually a very important and novel aspect of RECODE. In large-scale deep RL a key ingredient for the agent to learn a good policy is to receive a diverse stream of observations that are not strongly correlated to each other. This is usually achieved by instantiating multiple copies of the policy (i.e. multiple actors) that are used to *concurrently* generate multiple interaction trajectories. The diverse observations in these trajectories are again concurrently  fed to the agent to improve its policy.
> This mechanism is orthogonal to the memory mechanism, but once again having an appropriate degree of diversity stored in the memory usually leads to better learning performance.
>
> However many existing memory mechanisms (e.g., Badia et al. and Kapturowski et al.) rely on an episodic memory that is local to each instantiated policy/actor and reset at the end of each episode. This way the memory of one actor can never benefit from the information collected by another actor.
>
> Conversely, RECODE considers a global memory that is shared not only across multiple episodes but across all instantiated policy/actors. It is an interesting finding in and of itself that these diverse experiences do not create interference in the memory but rather improve learning performance.
>
> We will reformulate the passage to clarify that this is not a comment on distributed vs centralized agent architectures, but rather on how observations are collected and fed into the memory.

---

> ### Author Response · Authors · 2022-11-18
> **Reply to reviewer concerns [Part 2]**
>
> We  can now address the reviewer’s specific concerns.
>
> > Overall, the ideas are interesting but the paper is poorly written and has inadequate experiments. The experiments should include a clear comparison with the standard Markov assumptions of RL and the general methods included here.
>
> If the reviewer refers to running experiments in a Markovian environment, all of our experiments are in Markov environments (the game engines in Atari and DM-HARD-8 clearly store a minimal sufficient state to compute the next frame). All of the base RL agents used in our experiments are recurrent, but they operate under an assumption of partial observability of a Markov environment. If this does not answer your concerns, could you provide some clarification on your feedback?
>
> > Theoretically, the paper is muddled and confusing. The authors need to distinguish the standard RL assumption of Markov models, vs. the exploration length allowed. This is unclear in the paper.
>
> > The authors attempt to define "history" to incorporate exploration length allowed, but it is difficult to understand if this is actually the case.
>
> The notation that we introduce in section 2 is standard (see e.g. McCallum, 1995; Hutter, 2004; Hutter et al., 2009; Daswani et al., 2013), and a more general formulation of an MDP to have the Markov state do not depend only on the latest state but on an arbitrary subset of past history. This notation allows us to more flexibly accommodate recurrent agents (both MEME and VMPO base agents have a recurrent policy) or representations (BYOL-Explore uses a recurrent embedding function) whose policy/embedding do not depend only on the last observation but the last k. It is unclear to us what “the exploration length allowed” means in the context of your feedback. Could you please clarify?
>
> > The authors need to better clarify the distinctions between the contributions. They note "The first contribution of this paper is a general solution to (ii)." This is confusing. So does the "embedding function on observations or trajectories that encodes a meaningful notion of similarity" include and estimate of smoothed visitation counts?
>
> We will improve the exposition. In particular it is our first contribution (RECODE) which solves the problem (ii) of visitation count estimations given informative embeddings, while it is the second contribution (CASM) which solves the problem (i) of defining a meaningful notion of similarity in order to be able to define what a state is (a necessary prerequisite to start counting how many times it is visited). The "embedding function on observations or trajectories that encodes a meaningful notion of similarity" (e.g. CASM) embeds all past observations into a latent space. These latents are then passed to the online clustering of RECODE to extract the smoothed visitation counts using clustering.
>
> > Highly non-standard notion of "Interaction Process between an Agent and its Environment". As defined, the history is infinite. Most work presumes a Markov property to bound the history. This raises an instant computational problem. Is it not critical to bound the history for computational or practical reasons?
>
> The length of the history and the Markovianity of the model are two orthogonal concepts. Markovianity is a property of the environment, while the history notation is just a mathematical notation (and quite standard; see e.g. McCallum, 1995; Hutter, 2004; Hutter et al., 2009; Daswani et al., 2013). Sometimes the term “history” is replaced by the more restrictive “trajectory” (i.e. the history only up to the last episode reset if the domain is episode-based).
>
> This notation is designed to deal with general settings that include recurrent agents or recurrent representations. For example an environment might be non-Markovian w.r.t. the last observation, but still Markovian w.r.t. the last k observations and therefore the agent has to carry around a history containing the last five observations to learn well. This is the case in e.g. the MEME and VMPO base agents that we use.
>
> Note that our *notation* makes the embedding function take as input the whole history (for notational simplicity) but the embedder can disregard past observation and only keep around the last one (this is for example the case in AP/CASM but not BYOL-Explore). We agree that it is critical to bound the history for computational or practical reasons (we do exactly that in our algorithms), but we aimed to retain greater generality in our *notation* to cover a wider variety of agent architectures and representation learning methods.

---

> ### Author Response · Authors · 2022-11-18
> **Reply to reviewer concerns [Part 1]**
>
> As an introduction, we would like to expand on some standard nomenclature and core concepts that will help us answer the reviewer’s concerns.
>
> In our notation the history of an interaction process is the collection of all the interactions between an agent and an environment. This is just a notational tool (i.e. a mathematical formulation) and has no impact on the actual implementation of the agent or the environment.
>
> A Markovian process is an interaction process whose evolution does not depend on the whole history, but rather on a (potentially hidden) state that is sufficient to define all the future dynamics of the process.
>
> However in large environments (both real and simulated) this Markov state is commonly not exposed to the agent, which receive only a partial *observation* of it. This is again the case in our experiments.
>
> An environment that is Markovian w.r.t. a state might be non-Markovian w.r.t. the perceived observation that the agent receives of this state. However, it is often possible to extract a much better estimation of the state and recover Markovianity by looking not at one single observation but e.g., to the last k observations. This is once again the case in our experiments, since all the agents that we consider have a recurrent policy, and therefore aggregate information across multiple steps to decide how to act.
>
> Finally, both Markovianity and the history notation are orthogonal from the concept of memory as defined in our paper. Memory is a certain amount of information (a subset of the history, or some more compact statistic generated from it) that the agent carries around to make better decisions during its interaction with the environment. For example while chess is a fully Markovian game, chess agents carry with them memory of opening moves to help them play better, and can also use memory of previous board states to take better actions in the later stages of the game (for example by estimating the adversary’s strategy). The size (or length) of the memory is only loosely related to the history and the Markovianity of the state.

---

### Official Review · Reviewer_TrKY · 2022-10-24

**Confidence:** 4
**Correctness:** 3
**Technical Novelty And Significance:** 2
**Empirical Novelty And Significance:** 2
**Recommendation:** 3

**Clarity, Quality, Novelty And Reproducibility:**

The idea of using grouping together similar observations and keeping visitation counts for clusters is quite interesting but its novelty is limited.

Another weakness of this paper is that many of its parts are not presented clearly:
- First of all, it is not clear how the representation model is trained to compute the observations embeddings.
I found even Figure 2 a little bit confusing - Does c_t correspond to the cluster-center counts? It could be appreciated if authors can explain the representations learning process more clearly.
- Eq. 1 does not return the visitation count of an embedding $e$.
- Figure 10. referred to the main manuscript but it is presented in the supplementary material

According to the presented empirical results, the MEME algorithm outperforms RECODE algorithm in most of the examined Atari games. Why RECODE is preferable compared to the MEME algorithm?

**Strength And Weaknesses:**


**Strengths**

- The proposed RECODE algorithm can estimate a simple exploration bonus that can be considered by any RL agent
-  The idea of using an online clustering algorithm over learned representations of observations to estimate visitation counts for clusters of observations close to each other is quite interesting.
- Experiments have been conducted on hard-exploration tasks of Atari and DM-HARD-8 suits, showing that *RECODE* algorithm can explore the environment efficiently.

**Weaknesses**
- The idea of grouping similar observations and keeping visitation counts for clusters is not quite novel. The main novelty of this work can be considered to be the representation learning of from the observations.
- Some parts of the paper are not well presented and therefore hard to be followed by the reader. Specifically, Section 4 which describes the representations learning process should be revised carefully.
- It is not verified by the presented empirical results that the RECODE algorithm outperforms the MEME algorithm.

**Summary Of The Paper:**

This work proposes Robust Exploration via a Clustering-based Online Density Estimation (RECODE) algorithm. RECODE calculates an exploration bonus that any RL agent can use as an intrinsic reward signal to explore unknown environment areas. To compute intrinsic reward signal the visitation counts of observations is considered. For this purpose, an online clustering algorithm is applied over learned representations of observations that estimates visitation counts for clusters of observations that are close to each other.  Experiments have been conducted on hard-exploration tasks of Atari and DM-HARD-8 suits.


**Summary Of The Review:**

This work proposes an interesting idea to drive agents to explore unknown environment areas, but its novelty is incremental. Apart from that, some details are missed, and that makes it hard for the reader to understand all the details and reproduce the results if needed. Finally, the experiments do not validate the superiority of the  RECODE algorithm over the MEME algorithm in each examined environment.

---

> ### Author Response · Authors · 2022-11-18
> **Reply to reviewer concerns [Part 2]**
>
> > [Continued from Part 1]
>
> > It is not verified by the presented empirical results that the RECODE algorithm outperforms the MEME algorithm.
>
> We believe that, similarly to Rev. zKdb, part of this concern might stem from an insufficiently clear nomenclature for our baselines. Note that the exploration algorithms evaluated in the experiments are composed of three parts.
>
> - The first part is a base agent that oversees the overall RL learning process (e.g., executing actions and collection observations, computing adjusted returns, updating the policy, ...). As base agents we focus on MEME (Kapturowski et al. 2022), a recent improvement over Agent57 (Badia et al. 2020) that achieves much greater sample efficiency and is the current state-of-the-art on Atari, and a VMPO-based agent (see Guo et al. 2022) that is the current state-of-the-art on the DM-HARD-8 tasks.
> - The second part is an algorithm to construct intrinsic rewards. In addition to RECODE, we also consider the recent BYOL-Explore (Guo et al. 2022), NGU (Badia et al. 2020) and NGU's two building blocks, RND (Burda et al. 2018) and Episodic Memory (EMM, Badia et al. 2020).
> - The third part is a representation learning mechanism to generate observation embeddings that are fed to the intrinsic reward generator. We consider both Action Prediction (AP) and CASM embeddings. Note that some intrinsic reward algorithms cannot make effective use of the representation learning module (e.g., RND), while others merge both second and third part in a single approach (e.g., BYOL–Explore)
>
> For example, in our more detailed taxonomy the original MEME agent described by Kapturowski et al. (2022) is denoted as the MEME-NGU-AP baseline, and we compare it against our novel MEME-RECODE-AP agent where the only modifications is the changed exploration reward. We updated the experimental section to reflect this improved taxonomy, and added a table with more details on all combinations to App. B in the revised version. Using this clearer taxonomy it is much easier to communicate and see how the MEME-RECODE-* family of agents clearly outperforms the original MEME-NGU-AP agent.
>
> We believe that an important focus for evaluating our contributions should be on the more challenging DM-HARD-8 domain being that there remains a significant gap between SOTA and human performance, and where our MEME-RECODE-CASM agent clearly outperforms all baselines. In particular Fig. 6/7 show that it reliably solves 6 tasks, while the original MEME-NGU-AP agent solves only 1.
>
> In fact the original MEME-NGU-AP agent is a poor baseline for DM-HARD-8, as its RND-based long term memory cannot deal with complex partially observable environments. Instead we compare against VMPO-BYOL-Explore (see Fig. 7) which is a better baseline (and previous SOTA)  capable of solving 5 tasks. Note that our MEME-RECODE-CASM achieves SOTA solving 6 tasks and surpassing VMPO-BYOL-Explore (see also Fig. 14 in the revised version).
>
> To answer the reviewer’s concern more thoroughly we also added another baseline which uses the base MEME agent with NGU, but replaces AP with the CASM embedding. While this MEME-NGU-CASM baseline shows some clear improvement over the original MEME-NGU-AP baseline, solving 2 tasks consistently (and occasionally a 3rd), this is still much worse than our MEME-RECODE-CASM’s performance.
>
> The ATARI experiments are included as a more universally known benchmark for evaluating the performance of general agents, and for easier interpretability thanks to the simple characteristics of Pitfall and Montezuma Revenge. Again to address the reviewers concerns since submission we slightly expanded the hyperparameter sweep search on the RECODE-specific hyperparameters which resulted in improvements for our MEME-RECODE-AP agent, reported in the updated paper. It now clearly outperforms existing baselines, including to the best of our knowledge being the first agent to reach the end screen in Pitfall; while arguably being a much conceptually simpler agent than MEME-NGU-AP.

---

> ### Author Response · Authors · 2022-11-18
> **Reply to reviewer concerns [Part 1]**
>
> > The idea of grouping similar observations and keeping visitation counts for clusters is not quite novel. The main novelty of this work can be considered to be the representation learning of from the observations.
>
> While we agree that clustering states to estimate pseudo counts has received attention in the RL literature, we do not know of other works that concretely translate these approaches to large-scale deep RL as we do in our paper.
>
> To support this, our submission already includes an expanded discussion of “Related works” in Appendix B (Appendix C in the revised version of the paper).
>
> The first part (parametric methods) does not impact the novelty of RECODE, as it mostly deals with hashing techniques rather than explicit clustering. The second part (non-parametric methods) includes methods closer in spirit to RECODE, but despite some of these methods considering nearest-neighbors based approaches, none fully develop the idea of clustering to deal with the added complexities due to RL and representations which change over time; thereby injecting additional nonstationarity in the learning process. In contrast, this is a primary focus in our work.
>
> If the reviewer had any specific reference in mind that would like us to discuss we would be happy to do so.
>
> > Some parts of the paper are not well presented and therefore hard to be followed by the reader. Specifically, Section 4 which describes the representations learning process should be revised carefully.
>
> We revised Figure 2 to correct some mislabeled indices and replaced the “c_t” labels with “z_t” to avoid confusion with the counts, and now refer to these explicitly in the text. We’ve also added labels for the encoder and predictor.
>
> > It is not verified by the presented empirical results that the RECODE algorithm outperforms the MEME algorithm.
>
> We added aggregated performance graphs to the revised version (Appendix H) to clarify that our approaches achieve state-of-the-art results in both the Atari and DM-HARD-8 domains,  matching or outperforming the original MEME and BYOL-Explore algorithms. To measure aggregate performance we use average human normalized score (HNS) introduced by Mnih et al. 2015.
>
> On the more difficult DM-HARD-8 domain, already the current submission clearly shows our approach greatly outperforming the original MEME algorithm. Figure 6 clearly shows that our approach reliably solves 4 tasks, a very significant improvement over the original MEME baseline that can only unreliably solve 1. The best available baseline for this harder setting is actually BYOL-Explore, and in our new aggregated results (reported in Fig. 14) we clearly see that our approach outperforms the BYOL-Explore baseline, achieving both a higher mean capped Human Normalized Score and improved data-efficiency.
>
> On the ATARI domain (reported in Fig. 13) the left figure reports raw HNS values, and we see that at the end of the learning process our approach exceeds the MEME baseline.
>
> However above a score of 100 average HNS can be dominated by a particular game where the agent is largely superhuman (in this case Q*Bert). For this reason we also report average HNS but with each individual game’s score capped to 100 (i.e., matching human performance). This is equivalent to counting the percentage of games where the agent achieves at least human level.
>
> This is reported on the right side of Fig. 13, showing that our approach can solve all 8 hard exploration tasks to human level, matching the highly sophisticated original MEME agent (that was designed for this specific task) with a much simpler intrinsic reward mechanism. We are also adding to the supplementary material a video showing our agent reaching the end screen of Pitfall!; a feat that the original MEME agent could not achieve.

---

### Official Review · Reviewer_zKdb · 2022-10-25

**Confidence:** 3
**Correctness:** 3
**Technical Novelty And Significance:** 3
**Empirical Novelty And Significance:** 3
**Recommendation:** 3

**Clarity, Quality, Novelty And Reproducibility:**

Overall, the paper has some novel ideas, but it lacks some ablation studies that will really strengthen the paper.

**Strength And Weaknesses:**

**Strength**:

* Estimating the state visitation via clustering-based non-parameteric way is intuitive.
* The results in the figure show that the proposed method achieves similar performance as SOTA on several hard-exploration Atari games. When the tasks contain noisy observations, the proposed method surpasses the SOTA.

**Weaknesses**:

* The paper lacks an intuitive explanation of the detailed RECODE algorithm. There is barely any discussions on why the algorithm is designed as it is. It's not easy to understand when there is only a (rather complex) pseudocode block describing the algorithm.
* Similarly, in the experiment section, there is no comparison against other exploration algorithms or density estimation algorithms to show the effect of RECODE. The paper compares RECODE and MEME. But it would be more insightful to compare just on the exploration part of the algorithm. Does RECODE leads to different coverage or exploration pattern than other count-based or curiosity-based exploration methods?
* As for representation learning, it's important to do an ablation study on how good the proposed casual action-state masking method is compared to other representation learning methods such as contrastive learning on observation input, forward/inverse dynamics model learning, reconstruction on observation input, etc. When doing ablation studies, other factors (such as the exploration method) should be controlled to be the same.

**Summary Of The Paper:**

This paper proposes a new clustering-based non-parameteric way of estimating the state visitation density, and uses it to construct a reward to encourage more exploration of less-visited states. The paper also proposes a way of converting the state input into some embedding space. Overall, both ideas seem to have some novelties and merits despite the experiment section can be improved as detailed below.

**Summary Of The Review:**

The paper has a potential to demonstrate a novel algorithm. While it shows that it achieves a similar performance as the SOTA on atari games, it does not deliver enough ablation studies to help readers understand how good each component is, and what the new exploration algorithm brings, what the new insights are, why these design choices, etc.

---

> ### Author Response · Authors · 2022-11-18
> **Reply to reviewer concerns [Part 1]**
>
> >The paper lacks an intuitive explanation of the detailed RECODE algorithm. [...]
>
> We’ve revised the description of RECODE in Section 3 to make the algorithm more clear. Appendix D describes a connection to the DP-means algorithm which provides a theoretical grounding to our basic update scheme. The choice of removal strategy is not determined from this theoretical framework, but in Figure 8 we present empirical results to highlight qualitative differences in RECODE’s behavior when different removal strategies are used.
>
> >[...] in the experiment section, there is no comparison against other exploration algorithms or density estimation algorithms [...]
>
> We think there might be a bit of confusion in our algorithm taxonomy. In fact one of the central contributions of the experimental section is that we do demonstrate that RECODE outperforms several exploration baselines which differ only in the exploration part of the algorithm. With the exception of BYOL-Explore, which uses VMPO as the base agent; all other baselines use MEME as the base agent (with identical hyperparameters). These baselines include NGU (which is the exploration bonus used in the published MEME agent), RND, and Episodic Memory (EMM, also referred to as “NGU minus RND” in the original Badia et al. 2020 paper).
>
> To improve the exposition we adopted a new, clearer taxonomy (the new App. B in the revised version) and used it to clarify the experiments in Section 5. We report here a summary of the new explanation.
>
> Note that the exploration algorithms evaluated in the experiments are composed of three parts.
> - The first part is a base agent that oversees the overall RL learning process (e.g., executing actions and collection observations, computing adjusted returns, updating the policy, ...). As base agents we focus on (i) MEME (Kapturowski et al. 2022), a recent improvement over Agent57 (Badia et al. 2020) that achieves much greater sample efficiency and is the current state-of-the-art on Atari, and (ii) a VMPO agent (see Guo et al. 2022) that is the current state-of-the-art on the DM-HARD-8 tasks.
> - The second part is an algorithm to construct intrinsic rewards to guide the exploration process. In addition to  the proposed RECODE algorithm, we also consider the recent BYOL-Explore (Guo et al. 2022), NGU (Badia et al. 2020) and NGU's two building blocks in isolation, namely; RND (Burda et al. 2018) and Episodic Memory (EMM).
> - The third part is a representation learning mechanism to generate observation embeddings that are fed to the intrinsic reward generator. We consider both Action Prediction (AP) and CASM embeddings. Note that some intrinsic reward algorithms cannot make effective use of the representation learning module (e.g., RND), while others merge both second and third part in a single approach (e.g., BYOL–Explore)
>
> For example, in our more detailed taxonomy the original MEME agent described in (Kapturowski 2022) is denoted as the MEME-NGU-AP baseline, and we compare it against our novel MEME-RECODE-AP agent where the only modifications is the changed exploration reward. We added a table with more details on all combinations to App. B in the revised version.
>
> With this new notation it is easier to see how Figure 9 in the appendix (Figure 3 in the revised version) already contains an ablation w.r.t. the two main contributions of this paper (exploration mechanism and representation learning), as we compare the same base agent (MEME) with a variety of possible exploration/representation approaches including
> - RECODE reward with AP embeddings
> - NGU reward with AP embeddings
> - Episodic memory (EMM) reward with AP embeddings
> - original RND reward, which does not make use of representation learning
> - a natural new baseline based on RND rewards computed not on raw inputs but on concurrently learned AP embeddings
>
>
> RECODE matches or exceeds the performance of all of these possible exploration bonuses. Similarly, comparing across Fig. 6 and 7 (still Fig. 6 and 7 in the revised version) we can see that MEME-RECODE-AP is outperformed by MEME-RECODE-CASM on the DM-HARD-8 tasks, showing that simply by changing the representation we can improve performance. To address your concerns we also updated the revised version to include an additional ablation for our hardest domain (DM-HARD-8) in Fig. 7 that compares MEME-RECODE-CASM with MEME-NGU-CASM. Once again the RECODE reward outperforms existing approaches while keeping everything else (agent, representation learning) constant.
>
> As for the coverage pattern of RECODE, Figure 1 in the main paper and Figure 8 in the appendix are a partial attempt to characterize it and were instrumental to some of the design choices in RECODE.

---

> ### Author Response · Authors · 2022-11-18
> **Reply to reviewer concerns [Part 2]**
>
> >As for representation learning, it's important to do an ablation study  [...]
>
> We agree that it is important to explore this axis of ablation, and that is why in Fig. 6 and 7 we compare the same base agent and intrinsic reward both with AP and CASM (i.e. MEME-RECODE-AP and MEME-RECODE-CASM). During the rebuttal period we conducted additional experiments that are now included in the revision version, and a new comparison between MEME-NGU-CASM and MEME-NGU-AP shows once again that CASM provides superior performance.
>
> However in the limited amount of time afforded by the rebuttal window we could not complete the implementation effort of testing combinations of recode with other existing representation learning baselines. We will conduct additional experiments and include some more of these ablations (e.g. comparison with MEME-RECODE-BYOL-Explore) in the final version of the paper.

---

> ### Comment · Reviewer_zKdb · 2022-11-27
> **Thanks for authors' response**
>
> I would like to first thank the authors for their response. However, I don't find the revised version to have a better writing quality. There are many ways to address a specific problem. So it is important to motivate the proposed method well to convince other people that this is a good solution. For example, after reading the revised section 3, I still don't see why the authors make these specific design choices, what the motivations are. The authors fail to explain "why they do what they do", but focus too much on "what they do". Other reviewers share the same concern on the writing. I believe writing is as important to for a good paper. I suggest the authors revise the paper substantially and submit it to a future venue.

---

### Author Response · Authors · 2022-11-18
**Improvements in revised version thanks to general feedback**

We’d like to thank all of the reviewers for their feedback. We will address each of their concerns individually but at a higher level the main actions we’ve taken to improve the paper are as follows:

- Improving the naming convention for our proposed algorithm variants and baselines and adding a detailed taxonomy of all agents evaluated in the paper (Appendix B in the revised version): many of the concerns reported stem from the confusion w.r.t. where our contribution is novel vs which parts are taken from the literature. The revised version makes explicit that each agent used in our experiments is composed by i) base agent, ii) an intrinsic reward mechanism for exploration, and iii) a representation learning mechanism. Using this notation it is easier to show how RECODE and CASM achieve state-of-the-art performance, and makes it more clear how each ablation supports our claims.
- We added figures presenting aggregated performance metrics of our main results across both task suites (Appendix H in the revised version)
- We’ve revised many descriptions throughout the main text in an attempt to improve general clarity.
- We added a new version of Figure 2 (CASM architecture) which corrects some mislabeled indices, adds labels for f & g, and renames c_t -> z_t to avoid confusion with the cluster counts (and we now refer to z_t explicitly in the text)
- We’ve rerun our main Atari experiments with revised hyperparameters for the RECODE component, which improved overall performance. In particular our agent is now able to reach the end-screen in Pitfall; a feat which, to our knowledge, no other agent has previously achieved. (We include a video demonstrating this in the supplementary materials)
- We’ve added an ablation for NGU with CASM embeddings in the DM-Hard-8 domain
- Under the time constraints of the rebuttal window, it was not possible to include ablations which use BYOL as the representation learning method which feeds into RECODE, but we intend to include this in the final version of the paper.

---

### Decision · Program_Chairs · 2023-01-20

**Decision:**

Reject

**Justification For Why Not Higher Score:**

- Lack of clear justification for the proposed method
- Lack of clarity. The paper is not written well enough to justify acceptance.

I do feel that the paper is better than the reviewer scores indicate. I feel that it should be rated somewhere between 4 and 5. If the paper were rewritten and some additional work were put into the experiments, then I can see it being accepted at ICLR.

**Justification For Why Not Lower Score:**

N/A

**Metareview: Summary, Strengths And Weaknesses:**

The paper proposes an exploration bonus that can be used to push RL agents to explore previously underexplored states. This bonus uses online clustering together with representation learning to provide a kernel-smoothing-style estimate of the number of visits to states.  Experimental results are presented on Atari and DM-HARD-8 benchmark problems.

Strengths

- The problem is important

- Method improves significantly in experimental evaluations over Never Give Up (Badia et al. 2020b), another exploration bonus method, and its ablations

Weaknesses

- Lack of clarity. This was a significant barrier during the review period, as the exploration section did not clearly explain which method was which. Originally, the paper was so confusing that two reviewers independently mistook the experimental results as showing that the proposed method did not outperform baselines. During the rebuttal period the authors uploaded a revised version of the paper but after a close reading I feel that the paper's lack of clarity still presents a significant barrier to readers. Reviewer vYnp describes the paper accurately as "muddled and confusing."

- The paper's methodology is not well-justified. Individual steps are explained but it is hard to see how it fits together into a coherent whole.

As an example of the second weakness, the paper justifies its use of a soft kernel-based exploration bonus with the sentence, "However, when the state-space is very large or continuous, the exact visitation count is often uninformative since the same embedding may rarely be encountered twice."

In my view, the thing that makes exact visitation counts uninformative is if we can borrow statistical strength across related histories (or embeddings of histories). If the state space is large but we cannot borrow strength (e.g., if the state space is tabular) then exact visitation counts *are* informative, or at least are more informative than other kinds of counts. It is true that the same embedding may rarely be encountered twice --- this simply makes the problem hard and means that we need a very large number of samples.

As another example, what is it that makes an embedding good for use in estimating visitation? Some embeddings are much better than others but this is not clearly discussed.